# Poly-drug use among female and male commercial sex workers visiting a drop in centre in Mombasa County, Kenya

Kemunto Phyllys[1], Onesmus Wanje Ziro[1], George Kissinger[1], Moses Ngari[2,3], Nancy L. M. Budambula[4], Valentine Budambula[1] *

**1** Department of Environment and Health Sciences, Technical University of Mombasa, Mombasa, Kenya, **2** KEMRI-Wellcome Trust Research Programme, Kilifi, Kenya, **3** Department of Public Health, Pwani University, Kilifi, Kenya, **4** Department of Biological Sciences, University of Embu, Embu, Kenya

* valbudambula@gmail.com

**Data Availability Statement:** All data are in the manuscript and/or supporting information files.

## Abstract

The relationship between commercial sex work and drug use is complex and the two exacerbate each other. In Kenya, Mombasa County has one of the highest populations of drug users and commercial sex workers. Despite documentation of drug use among sex workers, most of the studies are based on self-reported history which is prone to social desirability and memory recall biases. It is in this context that we sought to establish actual drug use is this sub-population. A cross-sectional study was conducted to determine self-reported and confirmed drug use among 224 commercial sex workers accessing services at Mvita Drop-in. Actual drug use was determined qualitatively using 6 panel plus alcohol Saliva Test kit. The overall prevalence of self-reported and confirmed current use for at least one drug was 98% and 99% respectively. Regardless of the technique used, alcohol and tobacco products were the most consumed substances. Alcohol use increased significantly with age (P = 0.03). Risk of cigarette use and testing positive for cotinine was higher among those age 18 to 35 years compared to >35years at P = 0.001 and P = 0.002 respectively. Poly-drug use was common with 98% testing positive for more than one drug. The reason for drug use was sex work related pressure (88%) with 60% of the respondents reporting they cannot transact this business without drugs. Almost every commercial sex worker is a poly-drug user. We recommend targeted interventions for commercial sex workers.

## Introduction

Commercial sex refers to selling sex or purchasing of sex which includes but is not limited to exchanging sex for money, drugs, food, shelter, gifts or other items [1, 2]. Drug use among commercial sex workers (CSWs) is a public health problem and the two exacerbate each other. This is partly due to their shared epidemiological and environmental predictive factors. These factors include low socio-economic status, a history of child abuse, unstable housing, unemployment, frequenting entertainment venues and being single or separated [3–10]. People who use drugs (PWUDs) are likely to engage in sex for money, sex for drugs and sex for police

**Funding:** The authors received no specific funding for this work.

**Competing interests:** The authors have declared that no competing interests exist.

**Abbreviations:** ATS, Amphetamine-type stimulants; AUDIT-C, Alcohol Use Disorders Identification Test-Concise; CSW, Commercial sex workers; ERC, Ethics Review Committee; FSW, Female sex workers; MSM, Men who have sex with men; MSW, Male sex workers; PWUD, People who use drugs; USA, United States of America.

protection [11, 12]. On the other hand, sex workers are more likely to use drugs as they wait for clients, to gain courage to approach clients and to boost energy to serve clients [13–15]. In addition, they are likely to use opiates or depressants in order to relax after sex work or as a relief from associated trauma [15, 16].

In Mexico, most female sex workers (FSWs) use alcohol. One study reported 92% alcohol usage among respondents with 83% having met the Alcohol Use Disorders Identification Test-Concise (AUDIT-C) criteria for hazardous drinking. Factors that predicted hazardous drinking in this sub-population were any drug use in the past month, being a barmaid or dance hostess, alcohol use before or during sex with clients and working in a city with a higher marginalization index [17]. In a different study carried out in two United States of America (USA) and Mexico Border Cities Mexico, participants who entered sex work at age 15 or younger were significantly more likely to report voluntary or forced drug use within the first 30 days of entry [18]. In Tijuana, 61% of FSWs reported a lifetime ever use of methamphetamine while 38% were current users. In this city, smoking methamphetamine daily was associated with living in the red-light district, perceived homelessness, having a high income and age [19].

A study among Iranian at-risk women in Tehran reported sex work as the main source of income in almost half of the sample. In this study, 44.6% of respondents were opiate users while 55.4% used both opiates and methamphetamine [20]. Findings from a nationwide bio-behavioural survey in Iran reported history of drinking alcohol to be positively associated with lifetime history of drug injection and younger age at sex work debut [21].

In Kenya, the coastal region has had longer documented history of drug use. For example, by 2006 heroin was reported to have been available in the streets of Mombasa for over 25 years [22]. Mombasa County has one of the highest populations of drug users and commercial sex workers in Kenya [23, 24]. Mombasa being a cosmopolitan city and a tourist destination it creates an opportune environment for both drug use as well as sex work. Its proximity to neighbouring countries with porous borders makes it a convenient route for drugs on transit [25]. In Mombasa, FSWs reported a high prevalence of a lifetime ever use of alcohol (91%), khat (71%) and marijuana (34%) with a vast majority (79%) having practiced poly-drug use [26].

Even though drug use among sex workers is documented, the documentation process is based on self-reported history which is prone to social desirability bias and memory recall bias. It is for this reason that this study sought to determine drug use patterns based on self-reported history, actual drug use and social-demographic features associated with drug use among sex workers in Mombasa.

## Material and methods

### Ethics statement

The study was conducted according to the Helsinki declarations. The study was approved by Pwani University Ethics Review Committee (ERC/BSc/001/2018). Informed written consent was obtained from each participant. Participant assisted questionnaires and consent forms were prepared in both English and Swahili. No name was recorded on the participant assisted questionnaire or used anywhere in the study. Participants benefitted from health promotion messages on effects of drug use on their health and safe sex.

### Study design

A cross-sectional study conducted between February and April 2018.

## Study setting

The study was conducted at Mvita Drop in Service Centre, Mombasa County in Kenya. This centre is managed by a non-governmental organization that offers sexual reproductive services; HIV counseling, testing and care; and addiction counseling.

## Study participants and sample size determination

The study recruited both male and female sexual commercial workers aged above 18 years accessing services at the drop in Centre for a period not less than a month. The sample size was determined by use of the formula $n = Z^2pq/d^2$ as used by fisher and colleagues and cited in [27]. The p and q values were based on the findings of a similar study [27] that reported 79% of the used the participants were poly-drug users. To sum it all $n = Z^2pq/d^2$ translated to $n = 1.96^2$x 0.79 x 0.21/0.05 x 0.05 = 254.91 which was rounded off to 255. Upon obtaining written informed consent, 255 participants were enrolled using convenience and snowball sampling methods. However, thirty one (31) participants declined to provide a saliva sample and were excluded. In the end only 224 participants completed the process of study participation and their data analyzed.

## Variables and data measurement

Social demographic characteristics and self-reported drug use history were documented using a participant assisted questionnaire. Actual drug use was determined qualitatively using 6 panel plus alcohol Saliva Test kits as per manufacturer's instructions. These kits utilize monoclonal antibodies to detect high levels of illicit drugs in human oral fluids. We assessed use of cocaine, opiates, tetrahydrocannabinol, amphetamine, benzodiazepines, cotinine and alcohol.

## Statistical methods

Study participants characteristics and drug use were summarized as proportions. Only participants who reported ever using drugs and consented for saliva test were included (N reported in each table). Association of reported and saliva tested drugs with demographical features were tested using chi-square test of association or Fishers' exact test (where N per cell was <5) because the sample size was not powered to indefinite risk factors for individual drugs. Combinations of drugs used were presented using quadrilaterals Venn diagrams plotted using the *VennDiagram* package in R statistical software.

The agreement between the reported drugs use and those that tested positive by the saliva test was assessed using the Cohen's Kappa coefficient. The interpretation of the Kappa coefficient was adopted from the author's benchmark scale [28]. Using coefficient of agreement for nominal scales: <0 (poor); 0 to 0.2 (slight); 0.2 to 0.4 (fair); 0.4 to 0.6 (moderate); 0.6 to 0.8 (substantial); and 0.8 to 1.0 (almost perfect). Statistical analysis was performed using Stata version 15.1 (StataCorp, College Station, TX, USA) and R statistical software version 3.6.2.

# Results

## Socio-demographic characteristics of participants

A total of 224 commercial sex workers were successfully recruited and included in the analysis. This was a relatively youthful sample aged between 18 to 35 years (92%) and with majority (78%) being females. Half of the participants (50%) had secondary level education, 35% had attained tertiary level of education and only 15% had primary education. Approximately half of the participants were unemployed (53%), while 3.13%and 44% were in formal and informal employment respectively. One hundred and twenty eight (57%) participants were never

married and 76% were Christians. Nearly all the respondents (N = 222, 99.11%) had more than one sexual partners with 93.75% reporting three or more sexual partners. Use of condoms during sexual intercourse was common with 98% reporting having used a condom in their last sexual intercourse. The most commonly used condoms were the male condoms at 92.86% with only 7.14% of the participants reporting use of female condoms (Table 1).

## Commonly used drugs: Self-reported drug use

Of the 224 study participants, 220 (98%) reported current use of drugs while the remaining four (1.79%) were currently not using drugs. The six most common reported used drugs were;

**Table 1. Socio- demographic characteristics of commercial sex workers visiting a drop in centre in Mombasa, Kenya.**

| Demographics | N = 224 |
|---|---|
| Age in years–N (%) | |
| 18 to 35 | 207 (92.41) |
| >35 | 17 (7.59) |
| Sex–N (%) | |
| Male | 49 (21.88) |
| Female | 175 (78.13) |
| Education level–N (%) | |
| Primary school | 33 (14.73) |
| Secondary school | 111 (49.55) |
| Tertiary education | 80 (35.71) |
| Employment status–N (%) | |
| Formal employment | 7 (3.13) |
| Informal employment | 98 (43.75) |
| Unemployed | 119 (53.13) |
| Marital status–N (%) | |
| Never married | 128 (57.14) |
| Currently married | 9 (4.02) |
| Co-habiting | 32 (14.29) |
| Separated/divorced | 51 (22.77) |
| Widow | 4 (1.79) |
| Religion–N (%) | |
| Christianity | 171 (76.34) |
| Islam | 48 (21.43) |
| Hindu | 3 (1.34) |
| Others | 2 (0.89) |
| Current sexual partners–N (%) | |
| None | 2 (0.89) |
| One | 0 |
| Two | 12 (5.36) |
| Three + | 210 (93.75) |
| Condom use during last sexual intercourse–N (%) | |
| No | 5 (2.23) |
| Yes | 219 (97.77) |
| Type of condom used–N (%) | |
| Male | 208 (92.86) |
| Female | 16 (7.14) |

**Table 2. Self-reported drug use and saliva drug tests of commercial sex workers visiting a drop in centre in Mombasa, Kenya.**

| Reported drugs–N (%) | N = 220 |
|---|---|
| Alcohol | 157 (71) |
| Cigarette | 138 (63) |
| Shisha | 83 (38) |
| Khat | 68 (31) |
| Heroin | 62 (28) |
| Marijuana | 47 (21) |
| Cocaine | 16 (7.27) |
| Rohypnol | 16 (7.27) |
| Diazepam | 9 (4.09) |
| Cobblers glue | 3 (1.36) |
| Drugs tested positive–N (%) | N = 196 |
| Alcohol (ACL) | 134 (68) |
| Cotinine (COT) | 124 (63) |
| Opioids | 55 (28) |
| Tetrahydrocannabinol (THC) | 34 (17) |
| Amphetamine (AMP) | 31 (16) |
| Cocaine | 16 (8.16) |
| Benzodiazepine (BZO) | 15 (7.65) |

alcohol (71%), cigarette (63%), shisha (38%), khat (31%), heroin (28%) and marijuana at 21% (Table 2). In this sub-population poly-drug use was common with only 26% commercial sex workers reporting mono-drug use. In this study current use of two or more substances was reported as follows: two (21%); three (26%); four (11%); five (11%); six (1.28%); seven (1.29%); eight drugs (0.51%); and nine drugs (0.54%). The combinations of four most commonly used drugs were alcohol, cigarette, khat and shisha (Fig 1A). Of the 220 participants who reported drug use 8 (5.09%) reported concurrent use of alcohol, cigarette, shisha and khat (Fig 1A).

The proportion of CSWs aged above 35 years who reported alcohol (94%) use was significantly higher than those aged 18 to 35 years who reported 68% usage (Chi-square $P = 0.03$). The ratio of CSWs reporting cigarette use was higher among those age 18 to 35 years compared to >35years at 65% and 24% respectively (Chi-square $P = 0.001$). The percentage of men reporting use of heroin (Chi-square $P<0.001$) and other drugs (Chi-square $P<0.001$) was higher than among female CSWs. Reported heroin use was higher among CSWs who were formally employed (57%) than those in informal (18%) and unemployed at 34% (Fishers' Exact $P = 0.007$). Reported shisha use was higher in the Hindus (100%) compared to the Muslims (38%) and Christians at 35% (Fishers' Exact $P = 0.03$). The proportion of respondents who did not use condom during last sexual intercourse and reported heroin use was 80% and significantly higher than those who used condom at 26% (Fishers' Exact $P = 0.02$) as summarized in Table 3.

## Commonly used drugs: Confirmed drug use

Of the 224 study participants, 196 (88%) consented to the saliva test. Among the 196 tested, 194 (99%) tested positive for at least one of the drugs being assessed. The test results were as follows; alcohol (68%), cotinine (63%), opioids (28%), tetrahydrocannabinol (THC) at 17%, amphetamine (16%), cocaine (8.2%) and 7.7% for benzodiazepines (Table 2). The proportion of participants who tested positive for poly drug use was: two drugs (5.60%), three (9.23%),

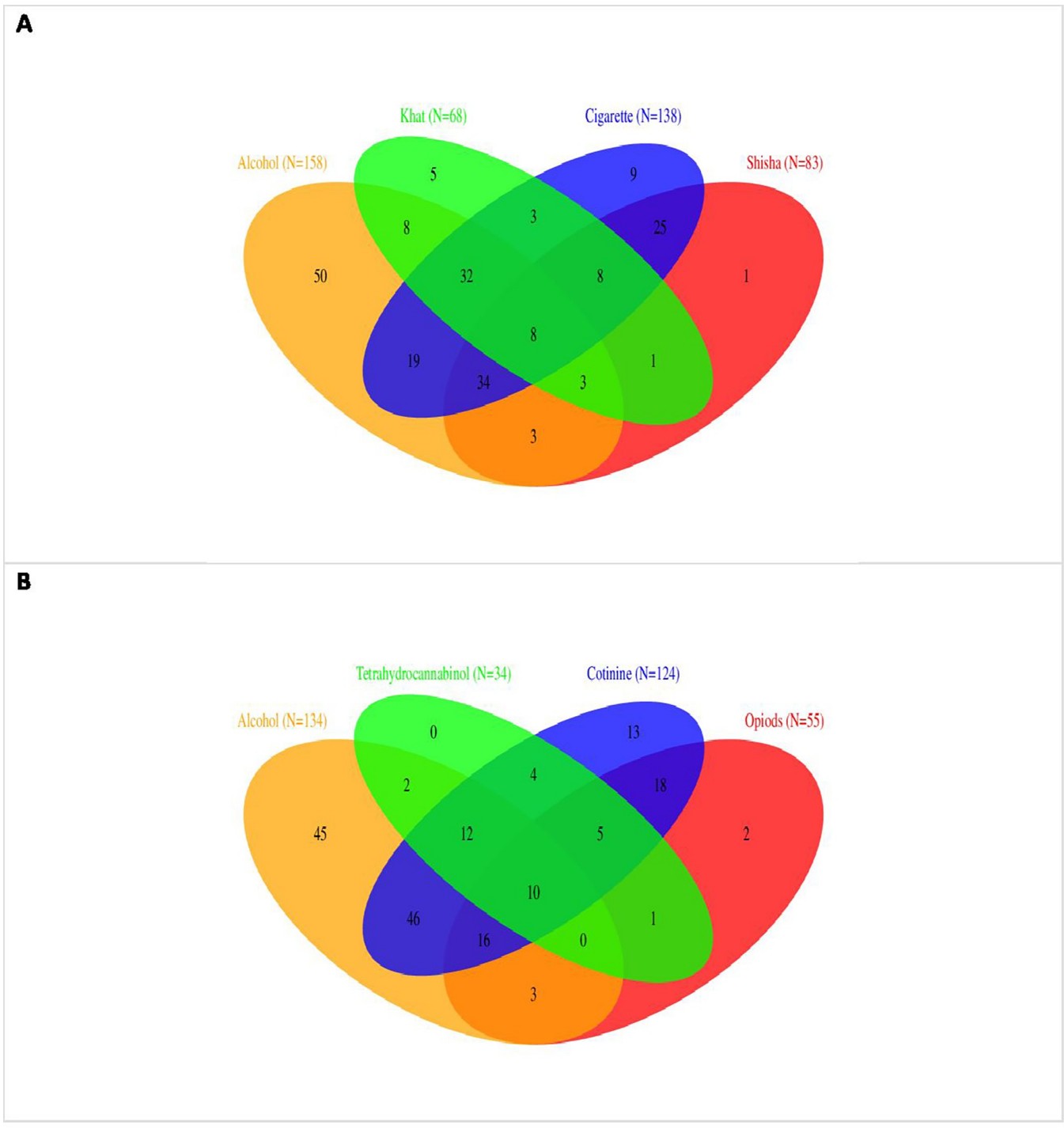

**Fig 1.** Venn diagram showing combinations of four most self-reported used poly-drugs (A) and drugs detected by saliva test (B) among transactional sex workers.

four (16%), five (31%), six (34%) and seven (1.52%) drugs. Among these positive participants only 2.02% were mono-drug users. The most popular combination was as alcohol, cotinine, THC and opioids (Fig 1B).

**Table 3. Self-reported drug use based on various demographics sub-groups among commercial sex workers visiting a drop in centre in Mombasa, Kenya.**

| Demographics–N (%) N = 220 | Alcohol N = 157 | Cigarette N = 138 | Shisha N = 83 | Khat N = 68 | Heroin N = 62 | Marijuana N = 47 | Others⁵ N = 29 |
|---|---|---|---|---|---|---|---|
| **Age in years** | | | | | | | |
| 18 to 35 | 141 (68) | 134 (65) | 80 (39) | 65 (31) | 59 (29) | 44 (21) | 27 (13) |
| >35 | **16 (94)*** | **4 (24)*** | 3 (18) | 3 (18) | 3 (18) | 3 (18) | 2 (12) |
| P-value# | **0.03** | **0.001** | 0.12 | 0.29 | 0.41 | 0.98 | 0.96 |
| **Sex** | | | | | | | |
| Male | 35 (71) | 32 (65) | 14 (29) | 18 (37) | 24 (49) | 15 (31) | **16 (33)*** |
| Female | 122 (70) | 106 (61) | 69 (39) | 50 (29) | **38 (22)*** | 32 (18) | 13 (7.43) |
| P-value | 0.81 | 0.55 | 0.16 | 0.27 | <0.001 | 0.06 | <0.001 |
| **Education level** | | | | | | | |
| Primary school | 20 (61) | 20 (61) | 13 (39) | 11 (33) | 8 (24) | 7 (21) | 1 (3.03) |
| Secondary school | 82 (74) | 64 (58) | 41 (37) | 28 (25) | 27 (24) | 24 (22) | 16 (14) |
| Tertiary education | 55 (69) | 54 (68) | 29 (36) | 29 (36) | 27 (34) | 16 (20) | 12 (15) |
| P-value | 0.33 | 0.38 | 0.95 | 0.24 | 0.32 | 0.96 | 0.17 |
| **Employment status** | | | | | | | |
| Formal employment | 4 (57) | 4 (57) | 3 (43) | 3 (43) | **4 (57)*** | 3 (43) | **4 (57)*** |
| Informal employment | 70 (71) | 59 (60) | 39 (40) | 30 (31) | 18 (18) | 18 (18) | 10 (10) |
| Unemployed | 83 (70) | 75 (63) | 41 (34) | 35 (29) | 40 (34) | 26 (22) | 15 (13) |
| P-values# | 0.69 | 0.85 | 0.68 | 0.74 | 0.007 | 0.26 | 0.009 |
| **Marital status** | | | | | | | |
| Never married | 90 (70) | 88 (69) | 50 (39) | 40 (31) | 33 (26) | 29 (23) | 19 (15) |
| Currently married | 6 (68) | 4 (44) | 4 (44) | 3 (33) | 3 (33) | 1 (11) | 1 (11) |
| Co-habiting | 21 (66) | 20 (63) | 14 (44) | 9 (28) | 6 (19) | 4 (13) | 0 |
| Separated/divorced | 38 (75) | 24 (47) | 14 (27) | 15 (29) | 19 (37) | 12 (24) | 8 (16) |
| Widow | 2 (50) | 2 (50) | 1 (25) | 1 (25) | 1 (25) | 1 (25) | 1 (25) |
| P-values# | 0.78 | 0.22 | 0.19 | 0.96 | 0.24 | 0.31 | 0.06 |
| **Religion** | | | | | | | |
| Christianity | 120 (70) | 101 (59) | **60 (35)*** | 48 (28) | 43 (25) | 36 (21) | 20 (12) |
| Islam | 34 (71) | 33 (69) | 18 (38) | 19 (40) | 17 (35) | 10 (21) | 8 (17) |
| Hindu | 3 (100) | 2 (67) | 3 (100) | 1 (33) | 1 (33) | 1 (33) | 1 (33) |
| Others | 0 | 2 (100) | 2 (100) | 0 | 1 (50) | 0 | 0 |
| P-values# | 0.54 | 0.20 | 0.03 | 0.23 | 0.16 | 0.57 | 0.35 |
| **Current sexual partners** | | | | | | | |
| None | 0 | 2 (100) | 2 (100) | 2 (100) | 2 (100) | 1 (50) | 1 (50) |
| Two | 9 (75) | 7 (58) | 4 (33) | 3 (25) | 3 (25) | 2 (17) | 3 (25) |
| Three + | 148 (70) | 129 (61) | 77 (37) | 63 (30) | 57 (27) | 44 (21) | 25 (12) |
| P-values# | 0.76 | 0.54 | 0.78 | 0.77 | 0.54 | 0.59 | 0.09 |
| **Condom use during last sexual intercourse** | | | | | | | |
| No | 3 (60) | 3 (60) | 3 (60) | 3 (60) | 4 (80) | 1 (20) | 2 (40) |
| Yes | 154 (70) | 135 (62) | 80 (37) | 65 (30) | 58 (26)* | 46 (21) | 27 (12) |
| P-values# | 0.64 | 0.98 | 0.36 | 0.17 | 0.02 | 0.99 | 0.13 |

*Significant difference (P<0.05), the p-values are from chi-square apart from # which are from Fisher exact test, ⁵includes Cocaine, Rohypnol, Diazepam& glue.

Among those CSWs whose saliva was tested, cotinine use was significantly higher among those aged 18 to 35 years (66%) compared to those aged above 35years at 25% (Fishers' Exact $P = 0.002$). Use of opioids was higher among those aged 18 to 35 years (29%) compared to those aged>35 years at 13% (Fishers' Exact $P = 0.03$). The percentage of participants using

cocaine and benzodiazepine was higher among male participants (20%) compared to the females at 6.90% (Fishers' Exact $P$ = 0.005). On the other hand, the proportional of CSW in formal employment using cocaine and benzodiazepine was significantly higher (50%) compared to those in informal (10%) and unemployed at 9.87% (Fishers' Exact $P$ = 0.04) as shown in Table 4. The use of drugs was not significantly different across the other demographic features (all P>0.05).

Among the 196 participants who reported current drug use and consented for saliva test, 117 (60%) reported use of alcohol and tested positive for alcohol. There was a moderate agreement between self-reported alcohol use and confirmed use at a Kappa coefficient of 0.53 (95% CI 0.40−0.66) (showing an agreement of 80.10%). Similarly, nine participants reported use of cocaine and tested positive for it at a Kappa coefficient of 0.64 (95%CI 0.43−0.86) (agreement of 95.4 3%) demonstrating substantial agreement between reported and confirmed cocaine use. Other drugs had good agreement; heroin/Opiods (almost perfect agreement, Kappa Coefficient 0.91 (95%CI 0.85−0.98)), Cigarette/cotinine (substantial agreement, Kappa Coefficient 0.79 (95%CI 0.70−0.88)), Khat/Amphetamine (moderate agreement, Kappa Coefficient 0.54 (95%CI 0.41−0.68)) and Marijuana/ Tetrahydrocannabinol (moderate agreement, Kappa Coefficient 0.48 (95%CI 0.32−0.64)) Table 5.

### Self-reported drug consumption patterns and practices

Among the 220 commercial workers who reported ever using drugs, over two thirds (84%) were initiated by friends while 43% and 0.94% were initiated by other commercial sex workers and siblings respectively. The frequency of drug use was daily for 94% of participants. The most frequent reported reason for drug use was sex work related pressure (88%). This was confirmed by the fact that 60% of the respondents reported they cannot be in this business without drugs. Surprising in this sub-population only 29% and 15% reported drug use due to peer pressure and family background. However, 40% were each using drugs because of drug or availability of money. More than three quarters (79%) of the participants had used drugs for more than three years. The respondents reported a range of effects after drug use including: anxiety (73%); feeling terrified (19%); hallucinations (19%); coughing (17%); and 14% experienced chest pains (Table 6).

### Discussion

A vast majority of the participants in the present study were female, youthful, unemployed or in informal unstable employment and single by the virtue of having never been married, separated, divorced or widowed. These findings were consistent with existing literature [10, 12, 16, 29–31]. Globally most sex workers are more likely to be women partly due social inequities which double as predictors of both CSW and drug use. For example in Eastern Nepal in India poverty was the most frequent (47.6%) factor leading to women engaging in sex trade [32]. It also is possible that the number of men who sell sex for money or goods (MSW) worldwide could be higher but underreported. Most studies and reports include MSW as either a subset of men who have sex with men (MSM) or as a subgroup of sex workers. Additionally, since this was a cross-sectional study it is not possible to know whether the separated, divorced or widowed participants joined CSW before or after marital separation.

In this study the whole sample reported attainment of formal education with the majority (85%) having attained post primary education. This deviates from most studies in Mombasa County and worldwide that report low levels of educational attainment among CSWs. For examples studies carried out from the same region in 2010 and 2016 reported 71% and 55% of theparticipants respectively had attained primary education or less [26, 33]. This could be

**Table 4. Saliva-positive drug use based on various demographics sub-groups among commercial sex workers visiting a drop in centre in Mombasa, Kenya.**

| Demographics–N (%) N = 196 | Alcohol N = 134 | Cotinine N = 124 | Opioids N = 55 | THC N = 34 | AMP N = 31 | Others[¶] N = 22 |
|---|---|---|---|---|---|---|
| **Age in years** | | | | | | |
| 18 to 35 | 121 (67) | 120 (66) | 53 (29) | 33 (18) | 30 (17) | 21 (10) |
| >35 | 13 (81) | **4 (25)**[*] | **2 (13)**[*] | 1 (6.33) | 1 (6.27) | 1 (5.91) |
| P-values[#] | 0.40 | **0.002** | **0.03** | 0.31 | 0.47 | 0.98 |
| **Sex** | | | | | | |
| Male | 24 (55) | 28 (64) | 21 (48) | 10 (23) | 7 (16) | 10 (20) |
| Female | 110 (72) | 96 (63) | 34 (22) | 24 (16) | 24 (16) | **12 (6.0)**[*] |
| P-values | 0.08 | 0.86 | 0.12 | 0.26 | 0.98 | **0.005** |
| **Education level** | | | | | | |
| Primary school | 18 (68) | 18 (67) | 6 (22) | 5 (19) | 7 (26) | 0 |
| Secondary school | 70 (71) | 57 (58) | 27 (27) | 20 (20) | 14 (14) | 14 (14) |
| Tertiary education | 46 (65) | 49 (69) | 22 (31) | 9 (13) | 10 (14) | 8 (11) |
| P-values | 0.83 | 0.42 | 0.79 | 0.48 | 0.30 | 0.10 |
| **Employment status** | | | | | | |
| Formal employment | 2 (33) | 4 (67) | 4 (67) | 0 | 4 (67) | **3 (50)**[*] |
| Informal employment | 59 (66) | 55 (61) | 19 (21) | 14 (16) | 15 (17) | 9 (10) |
| Unemployed | 73 (72) | 65 (64) | 32 (32) | 20 (20) | 12 (12) | 10 (9.87) |
| P-values[#] | 0.09 | 0.98 | 0.09 | 0.57 | 0.07 | 0.04 |
| **Marital status** | | | | | | |
| Never married | 73 (68) | 76 (71) | 29 (27) | 24 (22) | 15 (14) | 12 (11) |
| Currently married | 4 (50) | 4 (50) | 3 (38) | 1 (13) | 3 (38) | 2 (25) |
| Co-habiting | 23 (77) | 18 (60) | 7 (23) | 4 (13) | 7 (23) | 1 (3.3) |
| Separated/divorced | 32 (65) | 24 (49) | 15 (31) | 5 (10) | 5 (10) | 6 (12) |
| Widow | 2 (67) | 2 (67) | 1 (33) | 0 | 1 (33) | 1 (33) |
| P-values[#] | 0.64 | 0.23 | 0.94 | 0.37 | 0.12 | 0.18 |
| **Religion** | | | | | | |
| Christianity | 105 (70) | 93 (62) | 38 (25) | 30 (20) | 24 (16) | 16 (11) |
| Islam | 26 (62) | 27 (64) | 15 (36) | 3 (7.1) | 7 (17) | 6 (14) |
| Hindu | 2 (68) | 2 (68) | 1 (33) | 1 (33) | 0 | 0 |
| Others | 1 (50) | 2 (100) | 1 (50) | 0 | 0 | 0 |
| P-values[#] | 0.47 | 0.86 | 0.19 | 0.12 | 0.91 | 0.74 |
| **Current sexual partners** | | | | | | |
| None | 2 (100) | 2 (100) | 2 (100) | 0 | 1 (50) | 1 (50) |
| Two | 7 (64) | 7 (64) | 3 (27) | 0 | 2 (18) | 9 (82) |
| Three + | 125 (68) | 115 (63) | 50 (27) | 34 (18) | 28 (15) | 164 (90) |
| P-values[#] | 0.98 | 0.99 | 0.53 | 0.13 | 0.45 | 0.17 |
| **Condom use during last sexual intercourse** | | | | | | |
| No | 4 (80) | 4 (80) | 4 (80) | 1 (20) | 1 (20) | 3 (60) |
| Yes | 130 (68) | 120 (63) | **51 (27)**[*] | 33 (17) | 30 (16) | 171 (90) |
| P-values[#] | 0.99 | 0.66 | 0.02 | 0.98 | 0.96 | 0.10 |

[*]Significant difference (P<0.05), the p-values are from chi-square apart from [#] which are from Fisher exact test, [¶]includes Cocaine and Benzodiazepine.

attributed to exchequer funded free primary and subsidized secondary education from 2003 that has increased enrollment as well retention in schools. Mombasa County has also undergone rapid socio-economic growth steered by increase in number of tertiary educational institutions. Due to limited government funding at tertiary level, some college students may be

**Table 5. Extent of agreement/concordance between reported drugs and tested using the saliva samples.**

| Drug (N = 196) | N reported and positive by saliva test | Level of agreement | Kappa coefficient (95% CI) | P-value* | Strength of agreement |
|---|---|---|---|---|---|
| Alcohol | 117 | 80.10% | 0.53 (0.40–0.66) | <0.001 | Moderate |
| Cocaine | 9 | 95.43% | 0.64 (0.43–0.86) | <0.001 | Substantial |
| Cigarette/cotinine | 114 | 90.28% | 0.79 (0.70–0.88) | <0.001 | Substantial |
| Marijuana/ Tetrahydrocannabinol | 21 | 84.24% | 0.48 (0.32–0.64) | <0.001 | Moderate |
| Heroin/ Opioids | 50 | 96.38% | 0.91 (0.85–0.98) | <0.001 | Almost perfect |
| Khat/ Amphetamine | 28 | 83.67% | 0.54 (0.41–0.68) | <0.001 | Moderate |

*The P-value test the null hypothesis that the agreement was by chance.

**Table 6. Reported drugs consumption history and practices among commercial sex workers visiting a drop in centre in Mombasa, Kenya.**

| Drug use history and practices | N = 220 |
|---|---|
| Drug use initiator–N (%)* | |
| Friends | 185 (84) |
| Siblings | 2 (0.91) |
| Other transactional sex workers | 94 (43) |
| Frequency of drug use–N (%) | |
| Daily | 206 (94) |
| Weekly | 14 (6.36) |
| Time of month when drugs are used–N (%) | |
| Daily | 216 (98) |
| End month | 4 (1.82) |
| Reason for use–N (%)* | |
| Peer pressure | 64 (29) |
| Family background | 33 (15) |
| Drug availability | 88 (40) |
| Money availability | 89 (40) |
| Job pressure | 193 (88) |
| Effects experienced after drug use–N (%)* | |
| Anxiety | 161 (73) |
| Terrified | 41 (19) |
| Coughing | 38 (17) |
| Chest pain | 30 (14) |
| Hallucinations | 41 (19) |
| Impaired judgment | 11 (5.00) |
| Risky behaviors | 14 (6.36) |
| Sleepy | 16 (7.27) |
| Duration on drugs–N (%) | |
| One year | 8 (3.64) |
| Two years | 12 (5.45) |
| Three years | 26 (12) |
| Above three years | 174 (79) |
| Practice transaction sex without drugs–N (%) | |
| No | 133 (60) |
| Yes | 87 (40) |

*The total percentage is >100% because participants can have more than one response.

engaging in commercial sex to raise tuition fees and complement their upkeep. Alternatively, the deviation could be due to sampling method as the respondents were from one site.

The overall prevalence of self-reported and confirmed current use for at least one drug was 98% and 99% respectively. Regardless of the technique used to assess drug use, alcohol was the most consumed substance and most participants were poly-drug users. The risk of alcohol use increased significantly with age. Previously a study from the same region reported 91% life time ever use of alcohol among FSWs [26]. In Uganda, a study focusing on alcohol use within the context of sex work in Kampala city reported that clients to FSWs encouraged the later to consume alcohol [26]. An integrative review of global literature on alcohol use among female sex workers and male clients identified multilevel contexts of alcohol use in the sex work environment worldwide [13]. This high frequency of alcohol usage in this sub-population could be due to its effects as a depressant. Most CSWs are likely to use alcohol in order to gain courage to approach clients, engage in sex and to self-medicate traumatic episodes like sexual violence [15, 16, 34]. This was evidenced in this study by most participants (88%) who reported use of alcohol to cope with work related stress. In view of the fact that alcohol use among FSWs has also been identified as a contributor to risky sexual behavior [35] there is need to upscale targeted interventions for all CSWs, their clients and the social venues operators.

In this study, tobacco products were prevalently consumed with cigarettes and shisha being the most preferred substances. This was supported by the respondents testing positive for cotinine which is a metabolite of nicotine. Use of tobacco products however decreased with age. These findings were consistent with a survey that reported shisha use to be popular among the youth and commercial sex workers [36]. In Brazil, 71.1% of FSW were reported to be current smokers and smoking was associated with illicit drug use and alcohol consumption [37]. In Kenya, shisha has been projected in the public mind as trendy thus making it appealing to the youth who were the majority respondents in this study. This argument is strengthened with a survey that sought to establish the status of shisha and *kuber* use in Kenya [36]. The survey reported that shisha was popular among the youth and CSWs. Additionally, tobacco is a cash crop in Kenya and the usage of tobacco products other than shisha is not illegal but regulated [38]. This makes it easily accessible and affordable thus increasing its demand. Regardless of the existence of clear legal framework on tobacco use, the execution process has been feeble and slow.

The present study reports regular use of marijuana by FSWs in Mombasa. This was evidenced by self-reported current use (21%) or testing positive for tetrahydrocannabinol (17%). The findings deviate from a previous study from the same region that reported a 34% life time marijuana use [16]. This deviation can be attributed to the fact that this study focused on current use of marijuana while the previous study was reporting a lifetime use. These findings differ with results in South Africa that assessed depression, anxiety symptoms and substance use amongst sex workers. The study reported a very high (87.7%) lifetime prevalence of cannabis use among the respondents [16]. Overall marijuana use among FSWs was relatively low considering it is one of the most consumed illicit drugs in the world with over 188 million users [39]. Nevertheless, marijuana usage among CSW in Mombasa cannot be ignored. The practice can be attributed to the associated violent behaviour [40] which CSWs exhibit especially when dealing with clients who are unwilling to pay or abusive.

The respondents also commonly used khat which is structurally related to amphetamine. Cathinone is the main active ingredient in khat and its synthetic derivatives form a part of the new psychoactive substances list. These substances are collectively known amphetamine-type stimulants [41, 42]. In Phnom Penh, Cambodia, young female sex workers used amphetamine-type stimulants (ATS) to enable them to work longer hours; increase strength and endurance thus making possible women to serve more customers; and to change their

demeanour, making them more comfortable as well as friendlier with customers [42]. In Mombasa, a previous study among the FSW documented a prevalence of 71% khat use based on self-reported history [42]. This reduction in khat use could signify change in drug use patterns. Khat might have been replaced by more trendy drugs.

More than one quarter (28%) of the respondents reported current use of heroin and also tested positive for opioids. The risk of opiate use was associated with being younger, male gender, formal employment and having unprotected sex. Heroin, an opioid has been available in the streets Mombasa for over 40 years [43] and its use in Mombasa is progressively raising. For example, only six percent of FSWs in Mombasa reported a lifetime use of heroin in 2010 [42]. A study from the same region among injecting drug users (IDUs) reported a higher use of heroin among male IDUs. Additionally, a significant number of female IDUs reported engaging in unprotected sex [12]. On the other hand in Kisumu city, heroin use among FSWs was much higher (78%) although this was based on a relatively smaller sample size. In Kisumu, majority of the FSWs used heroin to engender morale and courage to engage in sex work as well as fight potentially abusive clients [12]. Use of heroin by CSW could be a coping strategy employed to deal with work related stress and physical trauma due to its analgesic properties.

A small proportion of participants used cocaine and its use was associated with being male and in formal employment. In the current study only 7.3% and 8.2% of the respondents reported usage and tested positive for cocaine respectively. Cocaine use among CSWs did not significantly increase with time as a previous study had reported 6% cocaine use among the FSWs [12]. This could be attributed to the fact that cocaine is a very expensive drug and is regarded as a high-end drug thus its demand is relatively low. Cocaine use before or during sex increases sexual desire at the same time it triggers sexually compulsive behaviours [44].

Current use of rohypnol and diazepam was also reported by a small proportion of participants who also tested positive for benzodiazepines. Divergent findings were reported in Pretoria, South Africa where a randomized controlled trial reported 1.5% and 4.2% current use of mandrax which is a methaqualone and rohypnol respectively among FSWs [45]. The usage of these tranquillizers and sedatives could be a form of self-meditation to relieve substance-induced depression or work related stress [46].

Based on self-reported history alcohol tobacco products and khat were frequently used; while saliva tests indicated alcohol, cotinine, THC and opioids to be the most preferred combinations. These results are in part similar to the findings of a study in Lagos state of Nigeria where19% of FSWs were dual alcohol and cigarettes users [47]. In South Africa, a study that sought to assess the association between poly-substance use and sex trade reported that 60% of the persons who tested positive for marijuana, cocaine and heroin compared to 37% of those who tested positive for marijuana only were involved in exchanging sex for drugs or money [48].

Peers in form of friends and colleagues played a major role in initiating participants into drugs use with a vast majority attributing drug use to work related pressure as they reported inability to transact business while sober. This could be the one of the contributors to the reported daily high frequency of drug use. Congruent to these results are findings of a study carried out in Kisumu city (Kenya). Participants attributed substance use to the need to gain courage in order to hold a conversation; ask for fair prices and demand condom use; and strength to contend with the potential for sexual assault [49].

## Conclusion

Having explored drug use among CSWs we conclude that drug use and use of multiple substances is indeed a problem in this sub-population. The overall prevalence of self- reported

and confirmed current use for at least one drug was 98% and 99% respectively. Regardless of the technique used to assess drug use, alcohol and tobacco products were the most consumed substances. Risk of alcohol use increased significantly with age while the odds of using tobacco products were correlated with younger age. We report poly drug use in this sub-population with 98% testing positive for more than one drug. Sex work related pressure and peer influence played a major role in initiation as well as continued use of drugs. A vast majority of the respondents reported that they cannot transact sex related business without drugs. Drug use among CSWs is therefore a complex problem that requires multi-dimensional interventions. We recommend targeted interventions for commercial sex workers. This may include economic empowerment through vocational training as well as user friendly harm reduction services.

The strength of this study is anchored on confirmed drug use using oral fluids which is more reliable than self-reported drug use. The first limitation is the cross-sectional design which precludes causal reasoning. Secondly all the sex workers were from one site and this could have introduced a selection bias. Finally, we tested for only seven drugs using a six (6) panel plus alcohol test kit. It is possible we might have missed to detect some drugs. Future studies may have to consider a wider panel like seventeen (17) plus alcohol test kit.

## Supporting information

**S1 Data.**
(CSV)

## Acknowledgments

We thank the study participants, management and staff of the Mvita Drop in Service Centre for their cooperation and support during the study.

## Author Contributions

**Conceptualization:** Kemunto Phyllys, Moses Ngari, Nancy L. M. Budambula, Valentine Budambula.

**Data curation:** Valentine Budambula.

**Formal analysis:** Moses Ngari, Valentine Budambula.

**Investigation:** Kemunto Phyllys, Onesmus Wanje Ziro.

**Methodology:** Kemunto Phyllys, Onesmus Wanje Ziro, George Kissinger, Moses Ngari, Nancy L. M. Budambula, Valentine Budambula.

**Software:** George Kissinger, Moses Ngari.

**Writing – original draft:** Kemunto Phyllys, Onesmus Wanje Ziro, George Kissinger, Valentine Budambula.

**Writing – review & editing:** Moses Ngari, Nancy L. M. Budambula.

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
