## [Decision Letter · Decision Letter 0]

21 Mar 2022

PGPH-D-22-00236

Poly-drug use among female and male commercial sex workers visiting a drop in centre in Mombasa County, Kenya

Dear Dr. Budambula,

Thank you for submitting your manuscript to PLOS Global Public Health. After careful consideration, we feel that it has merit but does not fully meet PLOS Global Public Health’s publication criteria as it currently stands. Therefore, we invite you to submit a revised version of the manuscript that addresses the points raised during the review process.

Please submit your revised manuscript by 30th April. If you will need more time than this to complete your revisions, please reply to this message or contact the journal office at globalpubhealth@plos.org. Please include the following items when submitting your revised manuscript:

We look forward to receiving your revised manuscript.

Kind regards,

Rubeena Zakar, Ph.D

Academic Editor

Journal Requirements:

1. Please amend your Financial Disclosure statement. If you did not receive any funding for this study, please simply state: “The authors received no specific funding for this work.”

2. Please update your Competing Interests statement. If you have no competing interests to declare, please state: “The authors have declared that no competing interests exist.”

3. In the online submission form, you indicated that “The data sets analyzed during the current study are available from the corresponding author on reasonable request.”. All PLOS journals now require all data underlying the findings described in their manuscript to be freely available to other researchers, either 1. In a public repository, 2. Within the manuscript itself, or 3. Uploaded as supplementary information.

4. Please provide separate figure files in .tif or .eps format only and remove any figures embedded in your manuscript file. Please ensure that all files are under our size limit of 20MB.

Additional Editor Comments (if provided):

Reviewers' comments:

Reviewer's Responses to Questions

**Comments to the Author**

1. Does this manuscript meet PLOS Global Public Health’s publication criteria? Is the manuscript technically sound, and do the data support the conclusions? The manuscript must describe methodologically and ethically rigorous research with conclusions that are appropriately drawn based on the data presented.

Reviewer #1: No

Reviewer #2: Yes

2. Has the statistical analysis been performed appropriately and rigorously?

Reviewer #1: No

Reviewer #2: No

3. Have the authors made all data underlying the findings in their manuscript fully available (please refer to the Data Availability Statement at the start of the manuscript PDF file)?

Reviewer #1: Yes

Reviewer #2: Yes

4. Is the manuscript presented in an intelligible fashion and written in standard English?

Reviewer #1: Yes

Reviewer #2: No

5. Review Comments to the Author

Reviewer #1: Dear Authors

Thank you for submitting the piece of scientific work, and especially, to come up with the real data that depicted the convetional self-reported one. It is also praiseworthy that the nexus between drug use and traded sex work has been described with empirical approach. Language is good. However, Some of the important recommendations to revise are the following:

It was found difficult to review, when there is neither page no. nor any line numbers, so pls assign pagination and line numbers so it would be easier for the reviewers. I am uploading the commented file along with it, for the easiness.

1. The manuscript is not in Vancouver style, pls (go through the link: https://journals.plos.org/plosone/s/submission-guidelines#loc-human-subjects-research)

2. In introduction last sentence, do you wish to check the hypothesis, if so, I could not find addressed in results section addressing it.

3. Sample size has been applied with FPC. I recommend to report the value of n/N, so that the reader could assess whether it is applicable or not?

4. Table 3 and 4, it is not clear that what the p-values signifies? I mean, the groups compared are not clear? So, better recommended to clarify and re-write.

5. In table, legend * is not clearly explained

6. Paragraphs, 3 and 5, in results section, the p-values reported and cited in corresponding tables do not possess the exact p-values, so clarify and recommended to re-write.

7. In same, paragraph 6, the kappa values were not found in any tables? Might be needed.

8. In discussion, 2nd paragraph (about the sex trade among college youth) and last paragraph (further recommendation of other drugs), recheck and clarify.

Reviewer #2: The discussion should be done to reflect the findings from the study. Need to look at relationships among the variables. Also there are quite some grammar errors especially at the discussion section. An example of these is the third sentence of the first paragraph of the discussion section. Some results were presented in the discussion section.

6. PLOS authors have the option to publish the peer review history of their article (what does this mean?). If published, this will include your full peer review and any attached files.

**Do you want your identity to be public for this peer review?** For information about this choice, including consent withdrawal, please see our Privacy Policy.

Reviewer #1: **Yes: **Chiranjivi Adhikari, Asst. Professor, Pokhara University, Nepal & PhD Scholar, Indian Institute of Public Health Gandhinagar (IIPHG), GJ, India

Reviewer #2: No

---

## [Decision Letter · Decision Letter 1]

23 Aug 2022

PGPH-D-22-00236R1

Poly-drug use among female and male commercial sex workers visiting a drop in centre in Mombasa County, Kenya

Dear Dr. Budambula,

Thank you for submitting your manuscript to PLOS Global Public Health. After careful consideration, we feel that it has merit but does not fully meet PLOS Global Public Health’s publication criteria as it currently stands. Therefore, we invite you to submit a revised version of the manuscript that addresses the points raised during the review process.

We look forward to receiving your revised manuscript.

Kind regards,

Joel Msafiri Francis, MD, MS, PhD

Academic Editor

Journal Requirements:

1. We have noticed that you have uploaded Supporting Information files, but you have not included a list of legends. Please add a full list of legends for your Supporting Information files after the references list. 

Additional Editor Comments (if provided):

Standardize the reporting of decimal points.

It would be helpful to adjust for sampling weights and clustering. The sample size estimation was misplaced as it did not account for design effect.

It would have been helpful to determine the sensitivity and specificity of self-reported drug use information against the saliva assays.

The analysis is descriptive – I wonder whether authors would be keen to carry out additional analyses. For example, regression?

Reviewers' comments:

Reviewer's Responses to Questions

**Comments to the Author**

1. If the authors have adequately addressed your comments raised in a previous round of review and you feel that this manuscript is now acceptable for publication, you may indicate that here to bypass the “Comments to the Author” section, enter your conflict of interest statement in the “Confidential to Editor” section, and submit your "Accept" recommendation.

Reviewer #1: All comments have been addressed

Reviewer #2: (No Response)

Reviewer #3: All comments have been addressed

2. Does this manuscript meet PLOS Global Public Health’s publication criteria? Is the manuscript technically sound, and do the data support the conclusions? The manuscript must describe methodologically and ethically rigorous research with conclusions that are appropriately drawn based on the data presented.

Reviewer #1: Yes

Reviewer #2: Yes

Reviewer #3: Yes

3. Has the statistical analysis been performed appropriately and rigorously?

Reviewer #1: Yes

Reviewer #2: Yes

Reviewer #3: Yes

4. Have the authors made all data underlying the findings in their manuscript fully available (please refer to the Data Availability Statement at the start of the manuscript PDF file)?

Reviewer #1: Yes

Reviewer #2: Yes

Reviewer #3: Yes

5. Is the manuscript presented in an intelligible fashion and written in standard English?

Reviewer #1: Yes

Reviewer #2: Yes

Reviewer #3: No

6. Review Comments to the Author

Reviewer #1: Since all the comments are addressed, I am recommending to the editor.

Reviewer #2: Please remove this from the discussion section because it is not part of the discussion "Some of the respondents reported to be currently using rohypnol (7.3%) and diazepam (4.1%). This was strengthened by 7.7% of the participants testing positive for benzodiazepines". The implication of the above results should be discussed.

Reviewer #3: Summary of the research

This is an interesting study on the burden of drug use among commercial sex workers visiting a drop-in center in Mombasa Kenya. In many similar studies, drug use is established by self-reporting whereas in this study they used an objective measure which adds strength. This is quite an important study as sex workers are among the key populations in HIV transmission and although strides have been made in reducing transmission, drug use among this group on top of worsening the health outcomes of individual sex workers, it has a ripple effect in increasing the risk of transmission of HIV and other STIs.

Overall, the authors did a good job in presenting the study findings in a clear and logical manner and the reviewer agrees with the authors that this study calls for targeted interventions towards commercial sex workers and incorporating drug screening in other harm reduction interventions so as to offer addiction counselling and rehab.

Minor issues

1. In the results section, line 204 where you report ‘Nearly all the respondents (98%) had concurrent……’, it is not correct. Recalculate

2. In Table 1: The decimal places are not given when reporting proportions in the brackets

3. In the results section line 211, it is not reported what happened to the 4 participants? Did they decline to answer the questions on drug use or they did not report any history of drug use? It is not clear

4. Fig 1A: You have reported ‘Miraa’ but in the text it is not mentioned anywhere, have that translated or use the standard term as in the text

5. In the result section, line 168 , line 273, line 363 have grammatical errors

7. PLOS authors have the option to publish the peer review history of their article (what does this mean?). If published, this will include your full peer review and any attached files.

**Do you want your identity to be public for this peer review?** For information about this choice, including consent withdrawal, please see our Privacy Policy.

Reviewer #1: **Yes: **Chiranjivi Adhikari

Reviewer #2: **Yes: **Tunde Adewale

Reviewer #3: No

---

## [Decision Letter · Decision Letter 2]

11 Oct 2022

Poly-drug use among female and male commercial sex workers visiting a drop in centre in Mombasa County, Kenya

PGPH-D-22-00236R2

Dear Dr Budambula,

We are pleased to inform you that your manuscript 'Poly-drug use among female and male commercial sex workers visiting a drop in centre in Mombasa County, Kenya' has been provisionally accepted for publication in PLOS Global Public Health.

Best regards,

Joel Msafiri Francis, MD, MS, PhD

Academic Editor

Reviewer Comments (if any, and for reference):

Reviewer's Responses to Questions

**Comments to the Author**

1. If the authors have adequately addressed your comments raised in a previous round of review and you feel that this manuscript is now acceptable for publication, you may indicate that here to bypass the “Comments to the Author” section, enter your conflict of interest statement in the “Confidential to Editor” section, and submit your "Accept" recommendation.

Reviewer #1: All comments have been addressed

Reviewer #2: All comments have been addressed

Reviewer #3: All comments have been addressed

2. Does this manuscript meet PLOS Global Public Health’s publication criteria? Is the manuscript technically sound, and do the data support the conclusions? The manuscript must describe methodologically and ethically rigorous research with conclusions that are appropriately drawn based on the data presented.

Reviewer #1: Yes

Reviewer #2: Yes

Reviewer #3: Yes

3. Has the statistical analysis been performed appropriately and rigorously?

Reviewer #1: Yes

Reviewer #2: Yes

Reviewer #3: Yes

4. Have the authors made all data underlying the findings in their manuscript fully available (please refer to the Data Availability Statement at the start of the manuscript PDF file)?

Reviewer #1: Yes

Reviewer #2: Yes

Reviewer #3: Yes

5. Is the manuscript presented in an intelligible fashion and written in standard English?

Reviewer #1: Yes

Reviewer #2: Yes

Reviewer #3: Yes

6. Review Comments to the Author

Reviewer #1: (No Response)

Reviewer #2: All my comments in the initial review have been addressed to my satisfaction.

Reviewer #3: The authors have addressed all the comments.

7. PLOS authors have the option to publish the peer review history of their article (what does this mean?). If published, this will include your full peer review and any attached files.

**Do you want your identity to be public for this peer review?** For information about this choice, including consent withdrawal, please see our Privacy Policy.

Reviewer #1: **Yes: **Chiranjivi Adhikari

Reviewer #2: **Yes: **Babatunde Adewale

Reviewer #3: **Yes: **Dr. Faith Aikaeli
